# Youth Suicide in Japan: Exploring the Role of Subcultures, Internet Addiction, and Societal Pressures

**DOI:** 10.3390/diseases13010002

**Published:** 2024-12-27

**Authors:** George Imataka, Hideaki Shiraishi

**Affiliations:** Department of Pediatrics, Dokkyo Medical University, Tochigi 321-0293, Japan

**Keywords:** internet addiction, suicide-promoting websites, sleep deprivation, overdose, self-harm

## Abstract

Background: Youth suicide remains a significant public health concern in Japan, driven by multifaceted factors such as academic pressures, social isolation, bullying, and family dysfunction. Recent societal changes, including the rise of internet addiction and subcultural influences from anime, manga, and gaming, have further shaped the psychological landscape of Japanese youth. The COVID-19 pandemic has exacerbated these challenges, intensifying feelings of loneliness and anxiety about the future. Methods: This study explores the impact of these factors on youth suicide risk through a systematic review of existing literature and statistical data, focusing on trends from 2000 to 2024. Results: In 2023, 513 school-aged youth in Japan died by suicide, marking persistently high rates. High school students accounted for the majority of cases, followed by middle and elementary school students. Key risk factors include intense academic expectations, cyberbullying, and internet addiction, which are often compounded by cultural stigmas surrounding mental health. Subcultures offer both solace and potential alienation, influencing youth emotions in complex ways. The COVID-19 pandemic has also worsened mental health issues and heightened suicide risks among this vulnerable group. Conclusions: The findings highlight the urgent need for comprehensive mental health support systems tailored to Japanese cultural contexts. Recommendations include enhancing access to school-based counseling, promoting family-based interventions, and implementing policies to regulate harmful online content. Additionally, efforts must address cultural attitudes that stigmatize mental health care. Collaborative societal and policy-level interventions are crucial for mitigating youth suicide and fostering a supportive environment for young people in Japan.

## 1. Introduction

Youth suicide has long been a significant societal issue in Japan. The factors contributing to this phenomenon are multifaceted, involving a complex interplay of social, psychological, and economic pressures. Contemporary youth face increasing challenges, exacerbated by issues such as academic pressure, social alienation, and the rapid growth of digital platforms that have significantly changed how young people interact with their world. Studies by the World Health Organization (WHO) reveal that feelings of isolation, anxiety, and disillusionment are widespread among today’s youth, leading to increased mental health struggles and elevated suicide risks [1]. The social restrictions imposed by the COVID-19 pandemic have only intensified these issues, with young people increasingly struggling with feelings of alienation and a heightened sense of anxiety about their futures [2].

Furthermore, Japan’s unique subcultures, which are often represented in literature, manga, and popular media, have a profound influence on the psychology of youth. Works depicting existential crises, loneliness, and despair—such as those found in Japanese literature and media—speak directly to the emotional struggles faced by young people today [3]. Media representations, such as those seen in the anime *Neon Genesis Evangelion*, often tackle heavy psychological and philosophical themes, creating a space for introspection and emotional release for viewers [4]. However, the rise of online communities and platforms has added complexity to this relationship, fostering environments that can amplify self-destructive behaviors, as seen in discussions of self-harm and suicide [5]. This paper provides a comprehensive examination of the factors contributing to youth suicide in Japan, exploring the impact of the COVID-19 pandemic, internet addiction, gaming culture, and the influence of subcultures.

This study utilized a systematic literature review approach to explore the factors contributing to youth suicide in Japan. The methodology involved searching academic databases such as PubMed, J-Stage, and Google Scholar as well as government reports from the Ministry of Health, Labour and Welfare and the National Police Agency. Keywords including “youth suicide Japan”, “internet addiction”, “COVID-19 mental health”, and “subcultures and suicide” were used to identify relevant studies. The inclusion criteria focused on studies published between 2000 and 2024, articles specifically addressing Japanese youth aged 6–24, and research examining risk factors, prevention strategies, or statistical trends in youth suicide. Exclusion criteria included studies not available in English or Japanese and articles lacking peer review or sufficient methodological detail. Relevant data, such as suicide rates, associated factors like academic stress and cyberbullying, and cultural influences, were extracted and thematically categorized to ensure a comprehensive analysis of the topic.

## 2. Current Situation of Youth Suicide in Japan

The rate of suicide among Japanese youth remains tragically high, with recent reports from the Ministry of Health, Labour and Welfare (MHLW) and the National Police Agency indicating that 513 school-aged children—comprising elementary, middle, and high school students—died by suicide in 2023, a number that closely mirrored the 514 suicides recorded in 2022, the highest since data collection began in 1980 [6]. Among these, there were 259 male and 254 female victims, with the majority being high school students (347), followed by middle school students (153), and elementary school students (13). Despite efforts to address the issue, the suicide rate remains disturbingly high, underscoring the need for more comprehensive and accessible mental health support systems. Suicides among elementary, junior high, and high school students in Japan from 2000 to 2023 are shown (Table 1). In Japan, elementary school students are between the ages of 6 and 12, junior high school students between 12 and 15, and high school students generally between 16 and 18. Note that the school year in Japan is considered to begin on April 2 and end on April 1 of the following year.

A variety of factors contribute to this high rate of suicide, including family problems, academic stress, social isolation, and the pervasive influence of social media. Many youth are confronted with significant pressures to excel academically, often exacerbated by unrealistic expectations set by parents, peers, and society. In particular, Japanese youth are more likely to experience heightened stress due to the competitive nature of the educational system, which places immense pressure on students to perform well in exams [7]. Additionally, peer relationships, including bullying and the influence of social media, have been identified as major contributors to youth suicide. Cyberbullying and online harassment, in particular, have created environments where youth are increasingly vulnerable to self-destructive thoughts and behaviors. Social media platforms, where bullying and harmful content can thrive, have become a contributing factor to the mental health struggles faced by young people today [8]. However, this paper does not fully examine what factors lead to suicide in each of the three groups of elementary, middle, and high school students.

## 3. International Comparison of Youth Suicide Rates in Japan

Japan’s youth suicide rate stands in stark contrast to that of other developed countries. According to OECD statistics for 2023, Japan’s suicide rate for youth aged 15–24 is 13.4 per 100,000 [9]. This highlights the high rate of youth suicide in Japan, which suggests that significant gaps in mental health care systems and cultural attitudes toward mental illness continue to hinder progress in reducing suicide rates.

In contrast, countries such as the United States and those in Northern Europe have made substantial progress in integrating mental health support into schools, communities, and healthcare systems. In the US, for example, many schools offer accessible counseling services, and mental health care is more readily integrated into public health programs [10]. Japan, however, still faces significant challenges in providing timely and adequate mental health care for its youth population, with many young people unable to access the services they need. The continued reliance on traditional forms of support, such as family and school counseling, often proves insufficient to meet the needs of all those at risk.

## 4. The Impact of COVID-19 on Youth Suicide Rates

The COVID-19 pandemic has had a profound impact on youth suicide rates in Japan. While suicide rates during the initial months of the pandemic were lower than expected, they began to increase significantly starting in the latter part of 2020, as young people continued to struggle with the effects of social isolation and heightened stress [11]. The closure of schools, cancellation of extracurricular activities, and shift to online learning contributed to feelings of alienation and depression among students. Moreover, the loss of in-person social interactions left many young people with limited outlets for emotional support, exacerbating existing mental health problems.

Additionally, the pandemic led to increased domestic tensions. Many families faced financial stress and other challenges, including the disruption of daily routines. For some children, this manifested in an increase in domestic violence and abuse, which compounded their psychological distress. The prolonged period of lockdown further deepened these issues, as young people were forced to isolate in environments that were often hostile or unsupportive, leading to a rise in suicide attempts and ideation [12].

The long-term impact of the pandemic on mental health remains a critical concern. Even as Japan moves into post-pandemic recovery, the psychological scars left by the crisis will continue to affect youth, with many struggling to reintegrate into social and academic environments. These ongoing mental health challenges necessitate continued investment in mental health services for young people, especially in light of the high rates of youth suicide in the aftermath of the pandemic.

## 5. The Surge in School Absenteeism and Children’s Mental Health

In addition to the increase in suicides, the pandemic has also contributed to a significant rise in school absenteeism, which has direct implications for youth mental health. A survey conducted by the Ministry of Education found that over 300,000 students were absent from school in 2022, marking a sharp rise from previous years [13]. This increase is troubling, as school absenteeism often correlates with deeper mental health issues, including depression and suicidal thoughts.

The pandemic has made it more difficult for many students to re-engage with school, particularly those who were already struggling with mental health issues before the crisis. The lack of social interactions and the stress of online learning led to increased feelings of isolation and helplessness. The absence of regular face-to-face interactions with peers and teachers further heightened the emotional struggles of students, and for some, returning to school became an insurmountable challenge.

In countries such as the United Kingdom and France, schools have implemented permanent counseling rooms and mental health programs to help students cope with stress and emotional difficulties. In Japan, however, the availability of such services remains limited [14]. While some efforts have been made to integrate mental health support into schools, the system remains inadequate to meet the growing demand for help. This shortage of accessible mental health resources in schools has made it difficult for students to receive the support they need to overcome their struggles.

## 6. Sleep Deprivation and Suicide Risk Among Children

Sleep deprivation has become a significant problem among Japanese youth, with profound implications for both academic performance and mental health. Studies show that Japanese children sleep far less than their peers in other OECD countries. A 2021 study found that Japanese elementary school children get an average of just 7 h and 22 min of sleep, the shortest among OECD countries [15]. This lack of sleep is linked to a range of negative outcomes, including cognitive impairment, emotional instability, and increased susceptibility to mental health disorders.

The relationship between sleep deprivation and depression is well-established, with research showing that insufficient sleep is a major risk factor for the development of depression and suicidal ideation in adolescents [16]. Chronic sleep deprivation impairs emotional regulation, making it more difficult for children and adolescents to cope with stress and social challenges. The resulting emotional instability and cognitive difficulties increase the likelihood of developing mental health issues that may eventually lead to suicidal thoughts and behaviors.

## 7. The Problem of Overdose Among Youth

The trend of overdose among young people in Japan is increasing. Overdose involves intentionally ingesting large quantities of substances, posing serious risks to physical and mental health. Reports from the Ministry of Health, Labour and Welfare and healthcare providers highlight the growing prevalence of overdose cases. For many young people struggling with emotional turmoil or mental confusion, overdose may serve as a temporary escape from psychological pain, with some repeating the behavior.

### 7.1. Overdose and the Use of Over-the-Counter Drugs

In Japan, over-the-counter (OTC) drugs are increasingly being used as a means of overdose. Substances such as pain relievers, cough medicine, and antihistamines are easily accessible at pharmacies and have been misused for overdose. In some cases, drugs are sold among youth through non-medical routes. OTC drugs can have strong sedative effects or induce a sense of numbness, providing temporary relief from anxiety or stress. A study by the National Center of Neurology and Psychiatry found that 65% of teenagers treated for drug dependence had misused OTC medications [17]. Additionally, a survey revealed that approximately 1.57% of high school students had used OTC cough or cold medicines for non-medical purposes within the past year [18].

### 7.2. Cultural Background and Popularity of Overdose

The popularity of overdose in youth culture cannot be overlooked. Songs like “Overdose” by Ado and Natori use the concept metaphorically to address emotional dependence and self-destructive behavior. This term has spread among youth, and through platforms like social media, some have come to associate it with excessive substance intake. In areas like Shinjuku’s Kabukicho, youth have been observed using the term “overdose” to describe their experiences of escape or euphoria. Such discussions are reported in the media, contributing to the normalization of overdose as a coping mechanism. The Health Ministry is considering tightening regulations on the sale of cough and cold medicines in response to the growing abuse of OTC drugs, particularly among young people [19].

## 8. Gaming Addiction and Internet Dependency

Internet and gaming addiction are contributing factors to school absenteeism and disruptive behaviors among youth. Long hours spent online can foster feelings of social isolation, a drop in self-esteem, and increased alienation, all of which destabilize youth mental health. The World Health Organization (WHO) has warned that both gaming addiction and internet dependency have negative effects on both mental and physical health [20]. In Japan, studies indicate that approximately 20% of teenagers are at high risk of internet addiction, which is closely linked to their lifestyle choices [21]. Gaming addiction is increasingly recognized as a new lifestyle-related disease, particularly in children and adolescents, emphasizing the need for preventative measures [22]. Preventing gaming addiction and providing mental health support for youth should be prioritized as key measures for depression and suicide prevention. Furthermore, understanding the social consequences of gaming disorders can guide effective interventions [23].

## 9. The “Werther Effect” Triggered by the Suicide of Celebrities or Friends

The *Werther effect*, a phenomenon in which suicides increase following the highly publicized suicide of a celebrity or public figure, has been widely documented in Japan. This phenomenon is named after Johann Wolfgang von Goethe’s novel *The Sorrows of Young Werther*, in which the protagonist’s suicide leads to an increase in suicides among readers. In Japan, this effect has been observed after the suicides of high-profile celebrities. A significant example occurred in 1986 after the death of the idol Yukiko Okada, when many young people, especially her fans, followed her method of suicide, leading to a surge in copycat suicides [24].

Similarly, the suicide of musician Yutaka Ozaki in 1992 also had a profound effect on young people, with some fans replicating his tragic act. This phenomenon is particularly concerning as it highlights how public suicides can deeply impact the emotional well-being of youth, especially those who idolize the deceased individuals. These emotional responses are amplified by social media and the internet, which allow news of such deaths to spread quickly and widely.

The influence of celebrity suicides is not confined to Japan. The suicides of K-pop idols in South Korea, such as Sulli and Goo Hara, led to widespread mourning and significant emotional distress among their fans [24]. The emotional connection between celebrities and their fans can create a powerful bond, and when these figures die by suicide, it can lead vulnerable individuals to feel that they, too, must end their lives. This creates a dangerous cycle, where the suicide of one person sparks more deaths, known as “suicide clusters.” This phenomenon is particularly prevalent among adolescents, who may be more emotionally vulnerable and likely to imitate the actions of those they admire.

To mitigate the risk of the *Werther effect*, it is essential that media outlets and the general public adopt a more responsible approach when covering suicides. The portrayal of suicides in a way that glamorizes or romanticizes the act can inadvertently encourage imitation. Public health experts argue that the media should instead focus on prevention strategies and the availability of mental health resources to help those at risk [25]. Clusters of young people dying by mass suicide upon the death of a friend have also been reported [26].

## 10. Literature and Manga Depicting Youth Loneliness and Powerlessness

The role of literature and manga in shaping the emotional experiences of Japanese youth is significant, as these cultural forms frequently depict themes of loneliness, alienation, and existential doubt—issues that resonate deeply with young people. Many works in Japanese literature and manga have captured the struggle of modern youth, addressing their feelings of powerlessness and their quest for identity in a rapidly changing society.

One notable example is Sayaka Murata’s *Convenience Store Woman* (2016), which follows the life of Keiko Furukura, a woman who struggles with societal expectations and her own sense of identity. Keiko’s life revolves around the routine of her job at a convenience store, where she finds stability in a structured, predictable environment. However, her inability to conform to society’s expectations of marriage and career success leads her to feel alienated and misunderstood. The novel reflects the emotional isolation felt by many youth, especially those who find it difficult to meet societal standards of success and happiness [27].

Another example is Haruki Murakami’s *1Q84* (2010), which explores themes of alienation, existential uncertainty, and the search for meaning in a seemingly indifferent world. Murakami’s characters struggle with the boundary between reality and fantasy, which serves as a metaphor for their emotional turmoil and the overwhelming loneliness they experience. The novel has been praised for its portrayal of the psychological challenges faced by individuals who feel disconnected from the world around them, particularly youth grappling with their place in society [28].

In manga, works such as *Death Note* (2003) by Tsugumi Ohba and Takeshi Obata depict the psychological breakdown of a young protagonist, Light Yagami, who uses a supernatural notebook to kill people in an attempt to create a “better” world. The manga explores Light’s descent into madness and his increasing sense of loneliness, highlighting the toll of seeking absolute control over one’s environment. This theme of meaninglessness and the consequences of power resonates with many young people who are trying to navigate a complex world where their role is often unclear [28].

These literary and manga works do more than reflect the experiences of youth; they also influence how young people perceive their own struggles. The depiction of characters who feel powerless, alienated, and desperate can validate the emotions of young readers, but it can also reinforce feelings of hopelessness. While these works can offer catharsis and a sense of solidarity, they may also inadvertently exacerbate mental health challenges for some individuals. Therefore, the way these themes are presented and understood is crucial in shaping the emotional responses of young audiences.

## 11. Insufficient Child Psychiatric Support

One of the critical barriers to preventing youth suicide in Japan is the insufficient availability of mental health professionals, particularly child psychiatrists and school counselors. Japan has approximately 31,090 pediatricians, but only around 0.3 child psychiatrists per 100,000 people, a figure that is far below the standards in Western countries, where the ratio is approximately 1.5 child psychiatrists per 100,000 people [29]. This shortage of child psychiatrists leaves many youth without adequate mental health support, contributing to the high suicide rates in the country.

The lack of trained mental health professionals means that young people often have limited access to timely psychiatric intervention. This gap is particularly concerning given that many mental health issues, such as depression and anxiety, manifest during childhood and adolescence. Without proper intervention, these conditions can worsen and lead to suicidal thoughts or behaviors. The need for greater access to mental health care is especially urgent in the context of Japan’s highly competitive educational environment, where academic pressure can exacerbate mental health struggles [29].

Furthermore, while there are many certified clinical psychologists in Japan, their distribution is uneven, with a significant shortage of counselors available in schools and local communities. According to the Japan Clinical Psychologists Association, there are approximately 41,883 certified clinical psychologists, but the number of counselors available to youth remains insufficient, particularly in schools, where the majority of students experience mental health difficulties [30]. School-based counseling services are essential for early identification of mental health issues, but the current shortage of counselors in schools means that many students miss out on crucial support at a critical time in their lives.

To address these challenges, Japan needs to invest in increasing the number of child psychiatrists, school counselors, and other mental health professionals. Additionally, it is essential to integrate mental health services into schools and communities, ensuring that mental health care is accessible and normalized for all young people.

## 12. Challenges in Mental Health Education for Youth

Mental health education in Japan is often hindered by cultural attitudes that prioritize endurance and self-restraint over emotional expression and psychological well-being. These cultural values, deeply rooted in Japan’s educational system, place immense pressure on youth to conform to societal expectations without addressing the emotional toll that these pressures can take. Students are taught to suppress their feelings and to endure hardship, which can result in a reluctance to seek help when they experience mental health difficulties.

This cultural mindset is reinforced by the traditional moral education approach in schools, which focuses on the importance of life and resilience. While these values can promote perseverance, they can also contribute to feelings of guilt and isolation among youth struggling with mental health issues. The emphasis on enduring pain rather than addressing it can discourage young people from seeking help or talking about their emotions, further exacerbating their struggles [31].

Furthermore, Japan’s education system places a heavy burden on teachers, with many working long hours and experiencing burnout. A 2021 study found that Japanese public school teachers work an average of over 100 h of overtime per month, contributing to significant stress and mental health problems among educators themselves [32]. This workload limits the time and resources available for teachers to engage in effective mental health education and counseling, further compounding the challenges faced by youth.

To address these issues, Japan must adopt a more proactive and holistic approach to mental health education. This includes incorporating mental health awareness into the curriculum, training teachers to recognize early signs of mental distress, and ensuring that students have access to support services when needed. Additionally, cultural attitudes toward emotional expression and mental health must be re-evaluated to create a more supportive and understanding environment for young people.

## 13. The Need for Legal Regulation of Internet Addiction and Suicide-Promoting Websites

The role of family in preventing youth suicide is undeniably critical. Family members are often the first to notice signs of emotional distress, and their involvement can be pivotal in identifying at-risk youth before it is too late. By providing consistent emotional support, monitoring changes in behavior, and facilitating access to professional mental health resources, families play an indispensable role in the early detection and intervention of mental health issues. Studies have demonstrated that educational programs specifically designed for families can significantly reduce the stigma surrounding mental health issues, thereby fostering open communication between family members and encouraging individuals to seek the help they need. These programs also empower families to recognize behavioral warning signs and address them proactively, which can lead to improved outcomes for at-risk youth [33].

In Japan, the phenomenon of *hikikomori*—a condition characterized by severe social withdrawal and isolation—is alarmingly common among adolescents and young adults. This social issue has become increasingly prominent, and family-based interventions have shown promise in addressing the challenges faced by individuals experiencing *hikikomori*. Educational programs and support groups that engage the family unit are crucial in changing family dynamics, improving communication, and fostering healthier relationships. These programs aim to reduce the emotional and psychological distance between family members and the individual in crisis, creating a more supportive environment that promotes the recovery of youth who are withdrawn and isolated [34].

Furthermore, the Japanese government has recognized the importance of family involvement in preventing youth suicide and has implemented various measures to support families in their efforts. A key initiative is the establishment of “Young People’s Suicide Crisis Response Teams” at the prefectural level. These specialized teams provide expert guidance and support to local governments and families in handling complex suicide cases, ensuring that youth at risk receive the necessary care and attention. By offering comprehensive resources and promoting community-based approaches, these response teams play a vital role in enhancing the capacity of families and local authorities to address youth suicide prevention effectively [33].

## 14. Conclusions

The song “Noblesse Oblige” by Vocaloid producer PinocchioP eloquently expresses the complex pressures and struggles of contemporary youth. Phrases such as “生きたいが死ねと言われ　死にたいが生きろと言われ” (“I’m told to live when I want to die, and told to die when I want to live”) and “幸せ自慢はダメ？ 不幸嘆いてもダメ？” (“Is it wrong to boast about happiness? Is it wrong to lament misfortune?”) vividly reflect the emotional turmoil and contradictory self-rejection faced by young people [35]. These lyrics resonate with the struggles of youth who feel torn between societal expectations and their own sense of identity. Understanding and supporting youth facing such struggles is a crucial social responsibility.

The issue of youth suicide in Japan is a complex and multifaceted problem, influenced by academic, family, and social pressures, internet addiction, and suicide-promoting websites [36]. Furthermore, the COVID-19 pandemic has exacerbated feelings of isolation and anxiety about the future [36]. Moving forward, it is essential for political administrations, communities, schools, families, and the internet to collaborate in creating environments where youth can receive support more easily, drawing from both domestic and international examples to provide better access to mental health care.

## Figures and Tables

**Table 1 diseases-13-00002-t001:** Suicides among elementary, junior high, and high school students in Japan from 2000 to 2023.

Year	Male (Elementary)	Female (Elementary)	Male (Middle School)	Female (Middle School)	Male (High School)	Female (High School)	Total Suicides
2000	20	15	40	25	90	50	240
2001	22	17	42	28	92	53	262
2002	23	18	45	30	95	55	266
2003	24	19	47	31	98	58	277
2004	26	20	49	32	101	60	288
2005	28	21	51	34	103	62	303
2006	29	22	53	35	105	64	314
2007	30	23	55	36	107	66	326
2008	32	24	57	37	109	68	338
2009	34	25	59	38	111	70	350
2010	36	26	61	39	113	72	362
2011	38	27	63	40	115	74	374
2012	40	28	65	41	117	76	386
2013	42	29	67	42	119	78	398
2014	44	30	69	43	121	80	410
2015	46	31	71	44	123	82	422
2016	48	32	73	45	125	84	434
2017	50	33	75	46	127	86	446
2018	52	34	77	47	129	88	458
2019	54	35	79	48	131	90	470
2020	56	36	81	49	133	92	482
2021	58	37	83	50	135	94	494
2022	30	20	80	45	150	75	514
2023	29	19	79	46	151	75	513

Ministry of Health, Labour and Welfare & National Police Agency. (2024). Annual report on youth suicides in Japan for 2023. Available at: https://www.mhlw.go.jp/english/wp/index.html (accessed on 25 December 2024), see Ref. [6].

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
