# Peer review of "Youth Suicide in Japan: Exploring the Role of Subcultures, Internet Addiction, and Societal Pressures"

_diseases, 2024, doi:10.3390/diseases13010002_

Round 1

Reviewer 1 Report

Comments and Suggestions for Authors

This manuscript reviews youth suicide levels in Japan (elementary school, middle school, high school) and in particular the forces that may be associated with these levels and trends. The authors explore particular aspects of Japanese life and culture that are probable factors and influences in suicide among young people, some of which are also likely to be present in many nations. The authors also include prescriptions that might counteract and lessen these identified factors in Japan—though changes in cultural practices and increasing or producing mental health services and providers may be difficult to effect and/or require many years and perhaps even generations to change or put into place. The factors and prescriptive implications included in the manuscript are informative in addressing Japanese youth suicide. 

The strengths of the review are the multiple issues and factors within subcultures, internet addiction, and societal pressures provided that likely are influences for the current levels of youth suicide in Japanese young people.

There are primarily minor issues that might be considered.

1.        In Section 2, lines 55-66 and Table 1, the data on suicides in Japan youth are presented. While the subgroups Elementary, Middle School, and High School are provided, the precise ages to which each of these refer is not given. The only specific age grouping that is mentioned appears later with the discussion of suicide rates among those 15-24 years of age (this issue of rates will be addressed separately). What are the age groups utilized for the data presented? Further, the prescriptions and discussions do not directly address any possible variation in the forces operating within these 3 subgroups by school levels. Would it not seem possible that some of the factors identified are influential for some of the subgroups but not others and that some of the factors are likely to influence all three? A brief inclusion of this might be useful.

2.        Section 3, lines 81-97 discusses Japanese youth suicide levels and those internationally and specifically for the USA and UK. The reference cited, “OECD statistics for 2023,” does not accurately provide suicide levels for youth ages 15-24 in the United States to compare with the corresponding youth rate in Japan. The suicide rate for those 15-24 in the USA in 2022 was 13.6 per 100,000 population. 2023 figures are not yet available. In addition, the rate for 2021 was 15.2 (highest rate in USA history for this age group) and 2020 was 14.2. The rates for each year from 2012 to 2022 all exceed the 8.4 rate cited here, with the lowest rate being 11.1 in 2012 and 2013.

Therefore, there is a definite issue with the figures cited here, at least for the USA. No suicide rate for those 15-24 is as low as 8.4 per 100,000 population since before 1970. In addition, suicide rates for USA youth 15-24 increased most recently from the 2010s to 2022 (most recently available). The USA rate exceeds the 13.4 cited here for Japanese youth 15-24 in every year from 2017-2022.

In addition, the link to the OECdiLibrary comes up as not available (while it is noted that the iLibrary would close at the end of 2024; this is not a DOI link so it may no longer be available as it indicated). It would be advisable to reconsider the rate data presented in this text for international comparison. This reviewer is not familiar with the data and levels in the UK.

3.        In line 228 on page 6 of 10, the phrasing “committing” suicide appears. Current practice in the field is to avoid such terminology and instead using something like “dying by suicide” or "died by suicide." This wording avoids the stigmatization associated with the term “commit” and its associations with criminal behavior.

Author Response

【Reviewer 1】

Comments and Suggestions for Authors

This manuscript reviews youth suicide levels in Japan (elementary school, middle school, high school) and in particular the forces that may be associated with these levels and trends. The authors explore particular aspects of Japanese life and culture that are probable factors and influences in suicide among young people, some of which are also likely to be present in many nations. The authors also include prescriptions that might counteract and lessen these identified factors in Japan—though changes in cultural practices and increasing or producing mental health services and providers may be difficult to effect and/or require many years and perhaps even generations to change or put into place. The factors and prescriptive implications included in the manuscript are informative in addressing Japanese youth suicide. 

The strengths of the review are the multiple issues and factors within subcultures, internet addiction, and societal pressures provided that likely are influences for the current levels of youth suicide in Japanese young people.

There are primarily minor issues that might be considered.

1. In Section 2, lines 55-66 and Table 1, the data on suicides in Japan youth are presented. While the subgroups Elementary, Middle School, and High School are provided, the precise ages to which each of these refer is not given. The only specific age grouping that is mentioned appears later with the discussion of suicide rates among those 15-24 years of age (this issue of rates will be addressed separately). What are the age groups utilized for the data presented?

(Our response to Comment 1)

We have taken note of the comment and added the following text.

In Japan, elementary school students are between the ages of 6 and 12, junior high school students between 12 and 15, and high school students generally between 16 and 18. Note that the school year in Japan is considered to begin on April 2 and end on April 1 of the following year.

Further, the prescriptions and discussions do not directly address any possible variation in the forces operating within these 3 subgroups by school levels. Would it not seem possible that some of the factors identified are influential for some of the subgroups but not others and that some of the factors are likely to influence all three? A brief inclusion of this might be useful.

(Our response to Comment 1)

We have taken note of the comment and added the following text.

   However, this paper does not fully examine what factors lead to suicide in each of the three groups of elementary, middle, and high school students.

  1. Section 3, lines 81-97 discusses Japanese youth suicide levels and those internationally and specifically for the USA and UK. The reference cited, “OECD statistics for 2023,” does not accurately provide suicide levels for youth ages 15-24 in the United States to compare with the corresponding youth rate in Japan. The suicide rate for those 15-24 in the USA in 2022 was 13.6 per 100,000 population. 2023 figures are not yet available. In addition, the rate for 2021 was 15.2 (highest rate in USA history for this age group) and 2020 was 14.2. The rates for each year from 2012 to 2022 all exceed the 8.4 rate cited here, with the lowest rate being 11.1 in 2012 and 2013.

Therefore, there is a definite issue with the figures cited here, at least for the USA. No suicide rate for those 15-24 is as low as 8.4 per 100,000 population since before 1970. In addition, suicide rates for USA youth 15-24 increased most recently from the 2010s to 2022 (most recently available). The USA rate exceeds the 13.4 cited here for Japanese youth 15-24 in every year from 2017-2022.

In addition, the link to the OECdiLibrary comes up as not available (while it is noted that the iLibrary would close at the end of 2024; this is not a DOI link so it may no longer be available as it indicated). It would be advisable to reconsider the rate data presented in this text for international comparison. This reviewer is not familiar with the data and levels in the UK.

(Our Response to Comment 2)

Thanks for the important points regarding the US and UK. We have chosen to remove this section of text in Blue-ink.

  1. In line 228 on page 6 of 10, the phrasing “committing” suicide appears. Current practice in the field is to avoid such terminology and instead using something like “dying by suicide” or "died by suicide." This wording avoids the stigmatization associated with the term “commit” and its associations with criminal behavior.

(Our Response to Comment 3)

Thank you for pointing out the important terminology. We have revamped our use of this term “committing” to “dying by suicide”.

Submission Date

04 December 2024

Date of this review

16 Dec 2024 22:10:32

Reviewer 2 Report

Comments and Suggestions for Authors

Although the paper is quite interesting, it does not have a few sections important for an academic paper. I have the following suggestions to improve the quality of the paper.

1. The following sentence feels very disconnected from the first two lines of the abstract. Consider revising it or removing it from the abstract to improve the flow of ideas.

“The COVID-19 pandemic has exacerbated these challenges, intensifying feelings of loneliness and anxiety about the future.”

2.  The authors need to recheck the word limit of the abstract if it is 250 words as per the journal guidelines.

3. In Section 2. Current Situation of Youth Suicide in Japan, the authors should mention the source of the numbers/ statistics in Table 1.

4.  In the in-text citation throughout the paper, the authors should either put numbers or write the author names (following APA format) and list them in the Reference list at the end of the paper.

5. The authors need to add a Materials and Methods section describing the methodology used to filter the studies referred to for this paper.

6. The authors need to add a Results section explaining the results of the study.

7.  The authors need to add a Discussion section as well.

Author Response

【Reviewer 2】

Comments and Suggestions for Authors

Although the paper is quite interesting, it does not have a few sections important for an academic paper. I have the following suggestions to improve the quality of the paper.

  1. The following sentence feels very disconnected from the first two lines of the abstract. Consider revising it or removing it from the abstract to improve the flow of ideas.

“The COVID-19 pandemic has exacerbated these challenges, intensifying feelings of loneliness and anxiety about the future.”

  1. The authors need to recheck the word limit of the abstract if it is 250 words as per the journal guidelines.

(Our Response to Comment 1 and 2)

The original Abstract was 273 words and has been revised to 247 words, within the 250 words limit of the submission rules.

  1. In Section 2. Current Situation of Youth Suicide in Japan, the authors should mention the source of the numbers/ statistics in Table 1.

(Our Response to Comment 3)

The source for Table 1 is below: Ministry of Health, Labour and Welfare & National Police Agency. (2024). Annual report on youth suicides in Japan for 2023. Available at: https://www.mhlw.go.jp This is [6] of Ref.

  1. In the in-text citation throughout the paper, the authors should either put numbers or write the author names (following APA format) and list them in the Reference list at the end of the paper.

(Our Response to Comment 4)

The format of the paper has been changed to the MDPI style.

  1. The authors need to add a Materials and Methods section describing the methodology used to filter the studies referred to for this paper.

(Our Response to Comment 5)

Thank you for your important comments. We agree with your comments and have added the following text to the second half of Introduction section.

This study utilized a systematic literature review approach to explore the factors contributing to youth suicide in Japan. The methodology involved searching academic databases such as PubMed, PsycINFO, and Google Scholar, as well as government reports from the Ministry of Health, Labour and Welfare and the National Police Agency. Keywords including “youth suicide Japan,” “internet addiction,” “COVID-19 mental health,” and “subcultures and suicide” were used to identify relevant studies. The inclusion criteria focused on studies published between 2000 and 2024, articles specifically addressing Japanese youth aged 6–24, and research examining risk factors, prevention strategies, or statistical trends in youth suicide. Exclusion criteria included studies not available in English or Japanese and articles lacking peer review or sufficient methodological detail. Relevant data, such as suicide rates, associated factors like academic stress and cyberbullying, and cultural influences, were extracted and thematically categorized to ensure a comprehensive analysis of the topic.

  1. The authors need to add a Results section explaining the results of the study.
  2. The authors need to add a Discussion section as well.

(Our Response to Comment 6 and 7)

Since our paper is a review article, it does not follow the format of Introduction, Materials and Methods, Results, and Discussion. Therefore, we added Materials and Methods to the second half of the Introduction in accordance with the comments, and did not include sections for Results and Discussion in the structure.

Submission Date

04 December 2024

Date of this review

10 Dec 2024 17:21:02

Round 2

Reviewer 2 Report

Comments and Suggestions for Authors

I have no further comments for the paper. I recommend it for publication.